# Research on Rural Population/Arable Land/Rural Settlements Association Model and Coordinated Development Path: A Case Analysis of the Yellow River Basin (Henan Section)

**DOI:** 10.3390/ijerph20053833

**Published:** 2023-02-21

**Authors:** Suxia Zhao, Mengmeng Yin

**Affiliations:** 1School of Surveying and Land Information Engineering, Henan Polytechnic University, Jiaozuo 454000, China; 2Research Centre of Arable Land Protection and Urban-Rural High-Quality Development of Yellow River Basin, Henan Polytechnic University, Jiaozuo 454000, China

**Keywords:** rural population, arable land, rural settlements, decoupling model, hot spot analysis, Yellow River Basin

## Abstract

The countryside is a complex regional system with population and land as the core elements, and it is of great significance to study the coordination of the rural human–land relationship for promoting rural ecological protection and high-quality development. The Yellow River Basin (Henan section) is an important grain-producing area with dense population, fertile soil, and rich water resources. Based on the rate of change index and Tapio decoupling model, this study took county-level administrative region as the evaluation unit to explore the characteristics of the spatio-temporal correlation model of rural population/arable land/rural settlements in the Yellow River Basin (Henan section) from 2009 to 2018 and the optimal path of coordinated development. The results show the following: (1) The decrease of rural population, the increase of arable land in a relatively large part of non-central cities, the decrease of arable land in central cities, and the general increase in the area of rural settlements are the most important characteristics of the Yellow River Basin (Henan section) for the change of rural population/arable land/rural settlements. (2) There are spatial agglomeration characteristics of rural population changes, arable land changes, and rural settlements changes. Areas with a high degree of change in arable land have a certain degree of spatial consistency with areas with a high degree of change in rural settlements. (3) The type of T3 (rural population and arable land)/T3 (rural population and rural settlement) is the most important temporal and spatial combination mode, and rural population outflow is serious. In general, the spatio-temporal correlation model of rural population/arable land/rural settlements in the eastern and western sections of the Yellow River Basin (Henan section) is better than that in the middle section. The research results are helpful to deeply understand the relationship between rural population and land in the period of rapid urbanization and can provide reference for the classification and sub-standard policies of rural revitalization. It is urgent to establish sustainable rural development strategies for improving the human–land relationship, narrowing the rural–urban disparity, innovating rural residential land area policies, and revitalizing the rural area.

## 1. Introduction

Human–land relations are the embodiment of the interaction between human activities and the natural environment, and the theory of human–land relations has been widely used to explain regional problems at different spatial and temporal scales [1,2]. Agriculture occupies about a third of the planet and employs more than 40% of the world’s workforce [3]. The relationship between human and land in the countryside is an existence that cannot be ignored. The rural regional system can be divided into population and social subsystems, economic subsystems, environmental subsystems, and resource subsystems, and the evolution of rural regional systems is a hot issue [4]. According to 2011 United Nations data, the rural population in the world’s least developed regions will increase to 3.12 billion by 2025 before decreasing to 2.87 billion in 2050. As a result, rural population growth will shift from positive to negative within 15 years, and rural demographic changes will have a huge impact on the global environment and social economy [5,6]. Population is often cited as a key driver of land occupation [7,8,9,10]. It is expected that the decline in the rural population will reduce pressure on the natural environment, but that depends on whether the population decline leads to changes in the use of rural land and the structure of land use, especially on the large proportion of arable land and rural settlements [11,12,13]. At present, research on the relationship between rural population and land is gradually enriched, and rural population flow between urban and rural areas is beginning to transform rural regional systems [14,15,16,17]. Accurately identifying the relationship between rural population and land is conducive to strengthening the combination of land use control and regional development positioning, conforming land use planning, and formulating customized rural development optimization paths.

Rural areas are the basic carriers of rural social economy. The development of economy and urbanization has promoted the rapid outflow of rural population, which has broken the original harmonious relationship between rural population and land, and the rural regional system is being reshaped and transformed [18,19,20,21,22]. Due to the long-term one-way migration of rural population to cities, the permanent rural population has dropped significantly [23,24,25]. However, the area of rural settlements has not decreased, but has gradually increased [15,26,27]. This threatens the arable land to some extent. The hollowing out of China’s rural areas and the multi-family foundation have led to idle and abandoned homesteads, and there is a widespread problem of a large amount of construction land encroaching on arable land [28,29,30,31,32]. This is a great waste of land resource utilization, urgently requiring coordination of the relationship between rural population and land to ensure sustainable development of rural areas.

The rural population–land relationship refers to the linkage and interaction of farmers, agriculture, and rural elements, which can be measured comprehensively through the change relationship of rural populations (farmers), arable land (agriculture), and settlements (countryside). Population is the main driving force of rural development, and population loss is the key factor causing rural decline, which is specifically manifested in the decline in the utilization efficiency of arable land and rural settlements [33]. Scholars have carried out a lot of research on rural populations and arable land and rural settlements, mainly in the following aspects: (1) Explore the driving mechanism of rural settlements change from the perspective of human–land interaction, and depict the theoretical process of rural settlements transformation [34,35,36]. (2) Study the coupling relationship and conversion relationship between arable land and rural settlements [37,38,39,40]. (3) Measurement of coupling degree and interaction relationship between rural settlements and rural permanent populations [6,34,41]. (4) The impact of rural population migration on the efficiency of arable land use. In addition, the research on the changes of rural human–land relations at different scales has attracted scholars’ attention to rural transformation and development, and the research on the transformation and development and interaction relationship between rural populations, land, and industry has gradually increased [42,43], which has clarified the direction for rural transformation and development and rural revitalization. Existing research on the relationship between rural populations and rural land mostly focus on the analysis of two factors, namely rural populations and agriculture, rural populations and rural settlements, rural settlements and arable land, and the lack of linkage analysis of rural populations, arable land, and settlements, respectively, representing the dynamic development of farmers, agriculture, and the rural area. On this basis, the lack of identification of the development combination of rural population/arable land and rural settlement has affected the future planning and reconstruction direction of rural areas to a certain extent.

The Yellow River Basin holds an indispensable position in maintaining food security and social and economic development, with the Henan section serving as a microcosm of rural development for the entire Yellow River Basin. It is an area with abundant water resources, high-quality arable land, and a concentrated population. Rural development faces multiple constraints, such as natural conditions, environmental protection, policy orientation, etc., and the income level of farmers is generally not high. Thus, the contradiction between humans and land is prominent. Environmental protection and high-quality development have become a national strategy [44,45]. There are currently several studies on the relationship between humans and land in rural areas of the Yellow River Basin. However, the human–land relationship in rural areas at the county level in the Henan section of the Yellow River Basin remains unclear. Exploring the coordinated pattern of the human–land relationship in county-level rural areas is conducive to promoting coordinated economic development and environmental protection. The optimization path for rural population/arable land/rural settlements can provide a differentiated theoretical basis for rural revitalization and sustainable development at the county level in the Yellow River Basin (Henan section).

Therefore, this paper aims to achieve the following: (1) A survey of temporal and spatial variation characteristics of the rural population/arable land/ rural settlements. (2) Make clear the coupling degree between rural population change and arable land change, and the coupling degree between rural population change and rural settlement change. (3) The temporal and spatial correlation model of rural populations, arable land, and rural settlements. (4) Aiming at the coordinated development of rural populations, arable land, and rural settlements, the differentiated regional development optimization path is proposed (Figure 1).

## 2. Materials and Methods

### 2.1. Overview of the Study Area

The Yellow River Basin (Henan section) is located in the transition zone known in China as the second to third ladders. It enters Henan through Sanmenxia and exits Puyang in the east, spread across the north-central part of Henan Province, with an area of 36,000 km^2^ and a total length of 711 km. It flows from west to east through 25 counties in eight prefectures and the cities of Sanmenxia, Jiyuan, Luoyang, Jiaozuo, Zhengzhou, Xinxiang, Kaifeng, and Puyang. The rural population of 25 counties in 2018 was 8.0106 million, accounting for 17.27% of the rural population of Henan Province. The arable land area is 1103.3 thousand hectare, accounting for 13.53% of the arable land area of Henan Province. The rural settlements area is 231.5 thousand hectares, accounting for 14.53% of the rural settlements area of Henan Province (Figure 2).

### 2.2. Date Source

The data in this study include the arable land area, rural settlements area, and rural population of 25 counties along the Yellow River Basin (Henan section). In order to maintain statistical consistency of the data, the arable land area and the rural settlements area in this paper are based on the results of the second Henan Provincial Land Survey in 2009, and the survey of annual land changes in Henan Province results in 2010–2018. Rural population data come from the 2010–2019 Henan Statistical Yearbook.

### 2.3. Research Methods

(1) Rate of change index. The statistical index population, arable land, and settlements change rate are used to express the degree of change and change trend of each index in the research base period.$\;L_{c}$ is the rate of change value for the population, arable land, and settlements, $S_{end}$ represents the index value of the last year; $S_{start}$ represents the index value of the first year, and the calculation formula is:$$L_{c}=\begin{array}{cc} \frac{S_{end}-S_{start}}{S_{start}} & \times 100\% \end{array}$$

(2) Tapio decoupling model. The term “decoupling” mainly comes from the field of physics and is generally interpreted as there being no codirectional response relationship between two variables. Since the OECD applied “decoupling” to the field of agricultural policy research, scholars have used the decoupling model as a way to measure the coupling relationship between economic growth and resource and environmental variables. In recent years, the decoupling model has been continuously applied to research on transportation, energy, arable land, water resource consumption, environmental pressure, etc. [46]. The relationship between the two factors can be studied through a ratio analysis of the rate of change between the two interrelated elements in a certain time period. The Tapio decoupling model is introduced to analyze the relationship between the rural population and arable land, as well as the rural population and rural settlements, and to explore the degree of coupling of changes between the rural population, arable land, and rural settlements. PE is the elastic index of change in the rural population and arable land (or decoupling elasticity index of the rural population and settlements change). CL is the rate of change of the arable land area or rural settlements area during the study period, and CP is the rate of change of the rural population. $L_{t}$ is the area of arable land or settlements in the last year of the study period, and $L_{0}$ is the area of arable land or settlements in the first year of the study period; $P_{t}$ is the number of rural populations in the last year of the study period, and $P_{0}$ is the population in the first year of the study period. The formula is as follows:$$PE=\frac{CL}{CP}=\frac{\left(L_{t}-L_{0}\right)/L_{0}}{\left(P_{t}-P_{0}\right)/P_{0}}$$

According to the PE elasticity index, there are eight decoupling types (Table 1).

## 3. Results

### 3.1. Changes of the Rural Population/Arable Land/Rural Settlements

The Yellow River Basin (Henan section) undertakes the functions of maintaining water and soil, providing grain, and regulating and controlling flooding, which is very important for national food security and protection of ecology and the environment. From 2009 to 2018, the arable land area in the 25 counties of the Yellow River Basin (Henan section) decreased, arable land decreased by 951 ha (Figure 3), the area of settlements showed an overall upward trend, the area of rural settlements increased by 14,240 ha (Figure 3), and the rural population gradually declined, displaying a total decrease of 1,723,400 people (Figure 4).

The map of rural population/arable land/rural settlements rate distribution in counties along the Yellow River Basin (Henan section) shows that the reduction of rural population is the main feature, and the distribution spatially forms subdivisions of varying degrees of population reduction. Arable land and rural settlements show two direction changes, as there are increases and decreases, and arable land and rural settlements have obvious areas with a high degree of change; the areas with a higher degree of arable land change are in the Jinshui District of Zhengzhou City and the Longting District of Kaifeng City, which is very consistent with the spatial distribution of areas with a high degree of change in settlements (Figure 5).

### 3.2. The Decoupling State of Rural Population/Arable Land and Rural Population/Rural Settlements

According to the decoupling model, the decoupling elasticity coefficients of rural population/settlements and rural population/arable land in counties of the Yellow River Basin (Henan section) were calculated, respectively. The rural population/arable land decoupling includes three types of decoupling combinations: strong negative decoupling (T3 population reducing and land increasing), weak negative decoupling (T4 rural population decreasing faster than arable land), and declining decoupling (T5 arable land decreasing faster than population) (Table 2, Figure 6). There are three combination types of rural population/rural settlements decoupling: strong negative decoupling (T3 population reducing and land increasing), weak negative decoupling (T4 rural population decreasing faster than settlements), and recession linking (T8 population reducing land at the same time, and decreasing synchronously). The decoupling types of rural population/arable land in the Yellow River Basin are mainly T3, while the rural population/rural settlements is mainly T4. (Table 3, Figure 6).

### 3.3. Decoupling Combinationof Rural Population/Arable Land/Rural Settlements

The combination analysis of the decoupling types of rural population/arable land and rural population/rural settlements changes can analyze the geographical relationship between the population/arable land/rural settlements. The combination of rural population/arable land decoupling and rural population/rural settlements decoupling has six modes: T3-T3, T3-T4, T4-T3, T4-T4, T5-T3, and T5-T8, of which rural population and arable land T3 and rural population and rural settlements T3 account for the largest proportion. The modes T5-T3, T5-T3, and T5-T8 account for the smallest proportion (Figure 7).

## 4. Discussion

### 4.1. Spatial and Temporal Patterns of Rural Population/Arable Land/Rural Settlements

The change law of rural population/arable land/rural settlements can effectively reveal the coordinated relationship between rural population and land in China and its differentiated performance patterns, which will be conducive to providing important support for rural land use transformation, and provide guidance for further promoting activities, such as intensive utilization and comprehensive improvement of rural land resources. There are significant spatial differences in the coordination status of population/arable land/rural settlements in the Yellow River Basin (Henan section), which is basically similar to the situation in the whole country [47,48].

Different rural population/arable land/rural settlements relationships directly affect and reflect the quality of rural development. This study finds that the permanent population decreased, the rural settlements increased, and the rural population/arable land/rural settlements changes were not coordinated in most areas of the Yellow River Basin (Henan section) from 2009 to 2018. Areas with different levels of economic development are also different in rural population changes, arable land changes, and rural settlements changes; areas with a higher degree of arable land change and areas with a higher degree of settlements change have a certain spatial consistency; areas with a higher degree of change are mainly concentrated in the central areas around the main urban areas of Kaifeng and Zhengzhou as areas with better economic development; the main problems focus on the contradiction between the reduction of arable land and the expansion of rural settlements; the economic-level development of the east and west is slightly inferior to the central region, and the prominent problem lies in the outflow of rural permanent population, where the area of rural settlements has not decreased but increased. The combination type of rural population/arable land decoupling is mainly T3, while the decoupling type of rural population and rural settlements belongs to T4.

### 4.2. Uncoordinated Change of Rural Population/Arable Land/Rural Settlements

This paper innovates the research from the coordination relationship between rural population, arable land, and rural settlements. Generally speaking, the reduction of rural population, the reduction of the area of rural settlements, and the increase of arable land are a relatively coordinated state of rural development under the current social development conditions. However, after research in this paper, it is found that the rural population/arable land/rural settlements in most areas of the Yellow River Basin (Henan section) are in a very uncoordinated state. The area of rural settlements has not decreased with the decrease of rural population but increased, the expansion of land used in rural settlements has occupied a large amount of arable land, and the area has also decreased.

At present, China’s urbanization has entered a transition period of rapid development, and there will be new changes in the flow of elements with the population as the core, which will bring about a new relationship of rural population/arable land/rural settlements changes. An in-depth understanding of this relationship is the basis for achieving the high-quality development of different types of villages. Economic development has strengthened the movement of rural population between urban and rural districts [23]. However, due to the restrictions of China’s land system, rural settlements can only be traded within the collective [48,49] and cannot be listed for free trade. Therefore, migrant workers entering cities have not really realized the transformation from rural to urban, and rural residents without settlements have built new settlements by occupying arable land, resulting in less arable land and increasing the area of rural settlements. It is worth noting that, in districts with different levels of economic development, the changes in rural population, arable land, and rural settlements are also different, which is related to urban agglomeration. Urbanization has led to the transfer of rural labor, capital, land, and other resource elements to cities [50,51,52,53]. Weak reverse decoupling (T4) is a rapid decline in the rural population rather than arable land. It is also a stage that China’s rural development must go through under the influence of urbanization. Eight districts belong to this category and are located in areas with rapid urbanization in the Yellow River Basin (Henan section). Strong reverse decoupling (T3) is the decrease of the rural population and the increase of arable land. During the period of rapid urban development, the area of arable land has also increased in many districts, which has a lot to do with China’s land policy, such as the policy of linking the increase or decrease of urban and rural construction land and the development of land consolidation work [54,55], which have promoted the increase of arable land area to varying degrees [56,57,58]. In addition, there are three districts belonging to the decline of the weak reverse decoupling (T4) of the population and land reduction, where the population decline rate is faster than the land decrease rate; this population outflow and settlement reduction phenomenon in some sense seems to belong to the coordinated change of population and land, but in fact, this is the result of the decline of rural vitality dominated by population reduction in the process of rapid urbanization development [59,60,61,62,63].

### 4.3. Driving Mechanism of Uncoordinated Development of Rural Population/Arable Land/Rural Settlements

The Yellow River Basin (Henan section) is a traditional plain agricultural area, which shoulders the important responsibility of ensuring national food security. The uncoordinated development of human–land relations in rural areas will lead to the emergence of food security problems and social problems. Through research, it is found that the current “population reduction and land increase” is the most important type of rural population/arable land/rural settlements changes in the Yellow River Basin (Henan section). The reason is that the affluent labor force is not supported by the corresponding level of industrial development, which has exacerbated the trend of young rural population outflow, which in turn has led to the emergence of rural diseases, such as the non-agriculturalization of land elements, the aging of the left-behind rural population, and the vacancy and abandonment of rural houses. However, the rural population, ploughing, and dwelling are interconnected and restricted, and their coordinated development is the foundation of rural development. For the current countryside of the Yellow River Basin (Henan section), the small number of people comprising the rural population will become a key factor restricting development. With the development of decades of reform and opening up, the education level of the population in rural areas has been continuously improved, and the quality of the population has been greatly improved. Land resources are the foundation for rural survival and development, and rural industrial development is also based on local resource endowment. Therefore, in order to realize the coordinated development of rural population/arable land/rural settlements, on the one hand, it is necessary to solve the problems of non-agricultural employment of the surplus rural labor force and the housing and education of urban migrant workers and their children, so as to ensure that the rural migrant population is “out of the ground and out of the countryside” and the scale of urban and rural amphibious population should be reduced. On the other hand, it should deepen the reform of the rural land system and actively explore appropriateness. In response to the exit model of rural homesteads with different regional environments and economic and social development, the coordinated development of rural human–land relations should be gradually promoted.

### 4.4. Optimization Path for Coordinated Development of Rural Population/Arable Land/Rural Settlements

Analysis and research of cold and hot spots and decoupling types of rural population/arable land/settlements in the Yellow River Basin (Henan section) found that the rural regional system of the Yellow River Basin (Henan section) mainly has the following problems: (1) The increase in the area of rural settlements does not match the large loss of rural population. (2) There are other land use types encroaching on arable land, so the differentiated regional development optimization path is proposed with the coordinated development of the rural population/arable land/rural settlements as the goal.

#### 4.4.1. Optimization Path of Arable Land Reduction Decoupling Combination Area

The decoupling combination type “T4-T3, T5-T3, T5-T8 rural population/arable land and rural population/rural settlements” should focus on coordinating arable land and rural settlements as well as other construction land. A large number of other land use types, such as construction land, encroach on arable land, and similar situations should be prevented from the following aspects. With regard to the remote suburbs of non-central urban and rural areas, we must gradually break down the dual structure of urban and rural areas and solve the problems of non-agricultural employment housing and children’s education of rural surplus labor. In addition, it must create conditions for rural migrants to “leave the land and leave their hometowns,” so as to promote the migration of rural non-agricultural employment population to cities and reduce the scale of rural settlements, thus reducing the erosion of arable land caused by the disorderly expansion of rural settlements. For the suburban rural areas of central cities, the scale of the original settlements should be strictly controlled with a view towards reducing arable land in Xingyang, Huiji, Jinshui, and other areas with a high level of urbanization. Though there is little likelihood of reclaiming arable land resources within the region, it should be linked to other non-central urban areas in the province to promote the increase and decrease of urban and rural construction land and stimulate the vitality of the improvement of idle land in non-central cities and villages, including (homesteads) and other construction land, while also increasing the total amount of arable land and strictly managing approval procedures for homesteads. Farmers should be urged to build their own houses and strengthen the protection of existing arable land in the outskirts of cities. Jiyuan, Yuanyang, and other areas with an average level of urbanization should promote the link between the increase and decrease of urban and rural construction land, while stimulating the vitality of rural idle land, including homesteads and other construction land, so as to increase the reserve resources of arable land.

#### 4.4.2. Optimization Path for Development of Rural Population Reduction and Combined Decoupling and Regional Development

The decoupling combination type “T3-T3, T4-T4 rural population/arable land and rural population/rural settlements” is located in most small cities of the county, the rural population migrates from small cities to big cities, which conforms to the current law of population migration.. In this regard, we propose the following measures. Since the outflow of the rural population around the small cities in the study area is still the trend for future development, increasing rural hollowing and improvement is key to solving the low efficiency of rural construction land utilization. The development positioning of each region should be clarified in the process of rural hollowing rectification. Relying on the existing towns, the model of new and expanded central villages and towns should be implemented to fully integrate rural construction land. The development of rural industries is a fundamental measure for reducing the outflow of rural population, and small cities and villages with serious rural population migration should consider their small or large cities, such as Zhengzhou, as markets for further optimizing infrastructure, increase policies to support the development of industries from which the population relocates, invest in rural infrastructure construction in a targeted manner, promote the upgrading of the agricultural structure and large-scale production of arable land, and optimize rural production/life/ecological space.

## 5. Conclusions and Prospects

### 5.1. Conclusions

The changing state of the rural population, arable land, and settlements profoundly affects the driving force and transformation of rural development. In this paper, we quantitatively analyze and identify changes in rural population/arable land/rural settlements, qualitatively identify the decoupling status of rural population/arable land and rural population/rural settlements, and identify the matching degree of the combination rural population/arable land and rural population/rural settlements. We also explore the degree of coordination of rural population/arable land/settlements in different regions and formulate a personalized rural population/arable land/settlements coordinated development path for the subregion of the Yellow River Basin (Henan section). The main conclusions are as follows:

(1) In 2009–2018, most counties along the Yellow River Basin (Henan section) showed a simultaneous decrease in rural population and arable land and an inverse synergistic situation of increasing land area of rural settlements, which was particularly obvious in the areas surrounding larger cities along the Yellow River Basin (Henan section). There were obvious characteristics of high and low value changes in rural population/arable land/settlements in the southern section of the Yellow River Basin. The low degree of population change near the central city corresponded to the high degree of arable land change area, and the rural vitality of the counties around the central city was better. The opposite was true for non-central urban and rural areas.

(2) During the study period, the changes in rural population and settlements along the Yellow River Basin (Henan section) were mainly strongly decoupled, resulting in an increasing per capita settlements area, while the actual settlements’ utilization efficiency decreased with the migration of a large number of rural populations, and there has been a fierce struggle between urban development and arable land protection. The changes in rural population and arable land are mainly based on strong negative decoupling and weak negative decoupling; arable land in most non-central cities and counties has increased and arable land in central counties has decreased, so surrounding cities need to share the pressure of protecting arable land in central urban areas while at the same time promoting the circulation of agricultural land in some non-central counties.

(3) There are six spatio-temporal combination models of rural population/arable land/settlements along the Yellow River Basin (Henan section), among which the combination type rural population/arable land “rural population decreases and arable land increases” and rural population/settlements “rural population decreases arable land increases” account for the largest proportion. This indicates that rural development is mostly in a period of natural decline in the face of rapid urbanization, and the loss of vitality of future rural development caused by serious rural population loss is a problem that must be faced in the future. Within this context, it is necessary to control the rate of rural population loss through rural industrial revitalization and appropriately guide rural population to return to the countryside for development as population are the source of power for rural revitalization.

(4) Most of the counties of the Yellow River Basin (Henan section) rural population-arable land/settlements are in a state of disharmony. “Rural population decreases settlements reduce arable land increase” and “rural population increases settlements decrease arable land increases” is the harmonious development type of rural population/arable land/settlements. Only Xin’an and Hubin currently meet these two types. With the outflow of the rural population, rural areas and rural settlements have been transformed or have begun to transform. There is a certain instability that makes it necessary to comprehensively regulate population and land elements during the transition period, seeking sustainable rural human–land relations, which are crucial to achieving rural revitalization at the population and land levels.

### 5.2. Limitations and Prospects of the Study

Under the huge differences in input and output of factors of production in urban and rural districts, the relationship between rural population and land tends to be increasingly fragmented. The relationship between rural population and land is facing the contradiction of declining and hollowing. The type of change and decoupling combination of the rural population/arable land/rural settlements in the Yellow River Basin (Henan section) proposed in this paper can more intuitively characterize the relationship between the three to a certain extent, better present the dynamics and potential problems of the rural population/arable land and rural settlements’ development, and clarify the direction for the optimization of rural human–land relations. However, the relationship between rural population and land is complex and phased. The impact mechanism of micro-scale farmers’ behavior and macro-scale policy orientation on rural development needs to be further explored. Longer-term continuous analysis will be the direction of future research.

## Figures and Tables

**Figure 1 ijerph-20-03833-f001:**
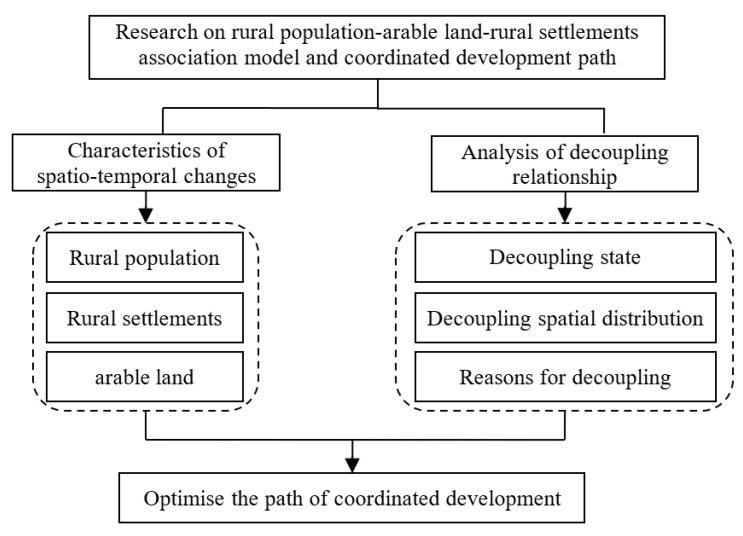
Theoretical framework diagram.

**Figure 2 ijerph-20-03833-f002:**
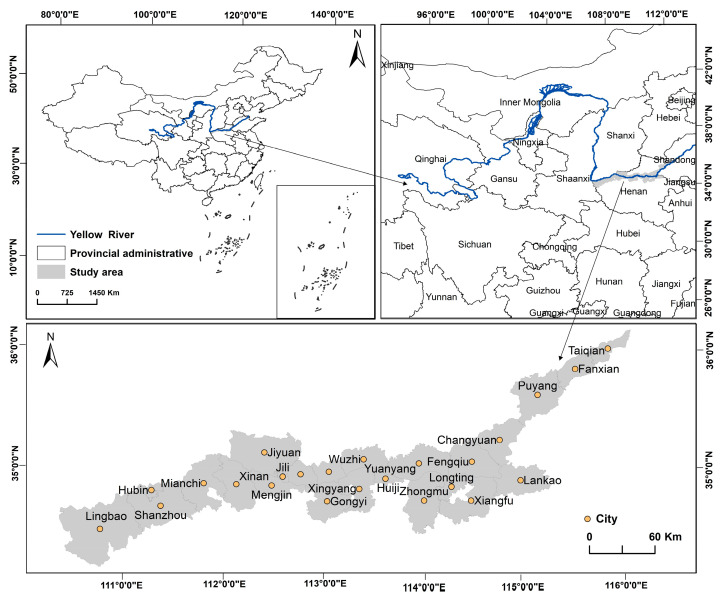
Study on the zoning map.

**Figure 3 ijerph-20-03833-f003:**
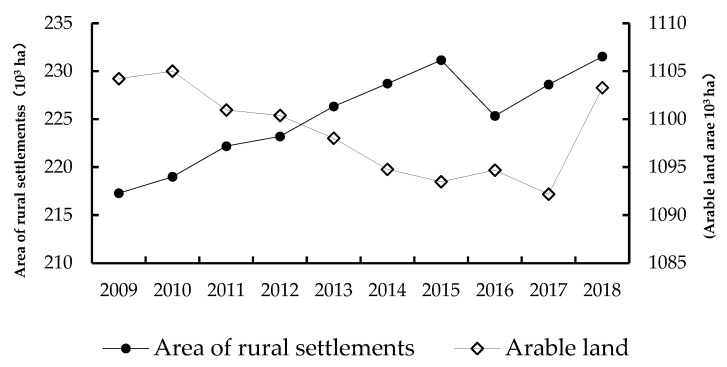
Changes of rural settlements land and arable land in the Yellow River Basin (Henan section) during 2009–2018.

**Figure 4 ijerph-20-03833-f004:**
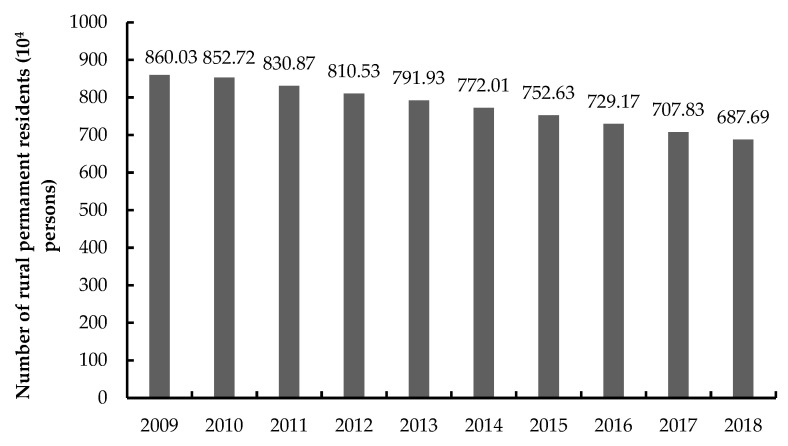
Changes of permanent rural population in the Yellow River Basin (Henan section) during 2009–2018.

**Figure 5 ijerph-20-03833-f005:**
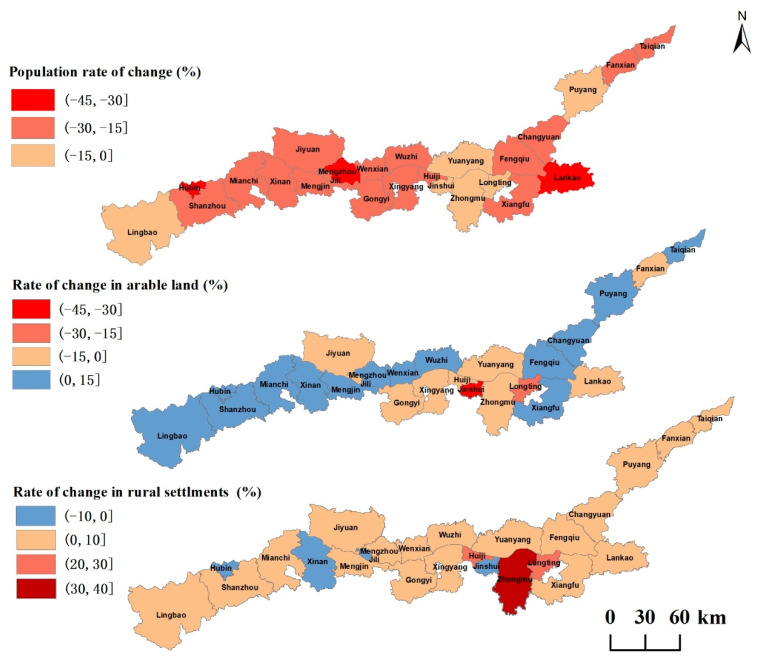
Distribution cold of and hot spots of changes of the rural population, rural arable land, and rural settlements in the Yellow River Basin (Henan section).

**Figure 6 ijerph-20-03833-f006:**
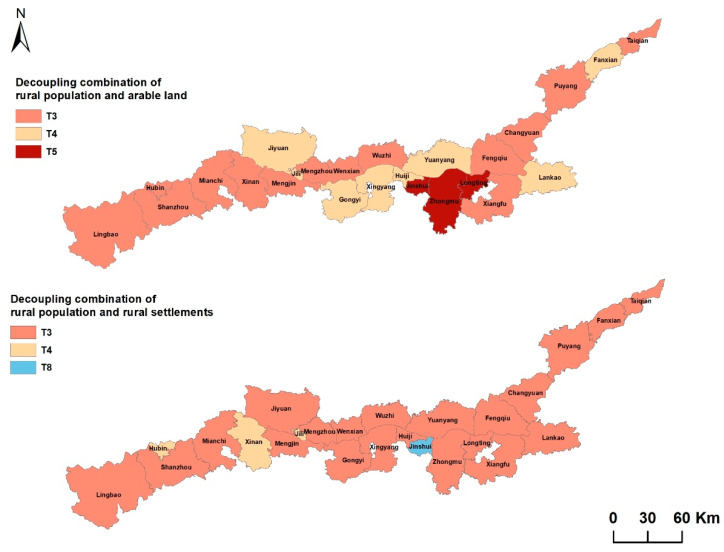
Distribution of decoupling types between permanent rural population and arable land between rural population and rural settlements in the Yellow River Basin (Henan section).

**Figure 7 ijerph-20-03833-f007:**
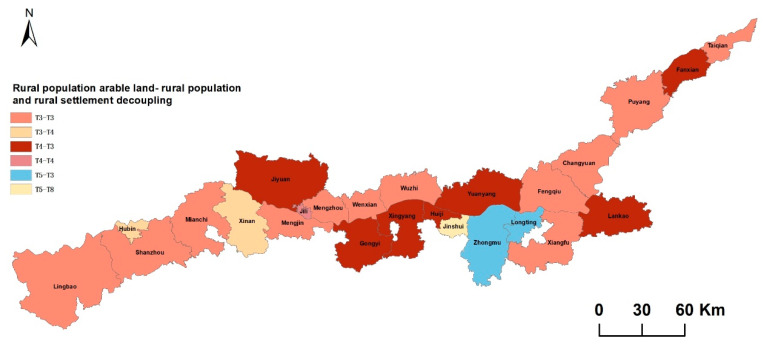
Combination decoupling types of the rural population, arable land, and rural settlements.

**Table 1 ijerph-20-03833-t001:** The rural population, arable land, and rural settlements decoupling state division.

The Decoupling State	Population Growth Rate	Arable Land/Settlements Growth Rate	Decoupling Elasticity Coefficient (PE)	State Interpretation
Weak decoupling (T1)	>0	>0	0 < PE < 0.8	The rural population and land have increased together, and the population has grown faster than the land.
Expansion inverse decoupling (T2)	>0	>0	PE > 1.2	Rural population and land have increased together, and land has grown faster than population.
Strong reverse decoupling (T3)	<0	>0	PE < 0	Land area has increased, and the rural population has decreased.
Weak reverse decoupling (T4)	<0	<0	0 < PE < 0.8	The rural population and land have decreased at the same time, and the population has declined rapidly.
Recession decoupling (T5)	<0	<0	PE > 1.2	The rural population and land have decreased at the same time, and the land has declined rapidly.
Strong decoupling (T6)	>0	<0	PE < 0	The rural population has increased, and the land area has decreased.
Expand the link (T7)	>0	>0	0.8 < PE < 1.2	The rural population and land increased at the same time, and the growth rate of the two was relatively synchronized.
Recession link (T8)	<0	<0	0.8 < PE < 1.2	The rural population and land have decreased at the same time, and the decline between the two has kept pace.

**Table 2 ijerph-20-03833-t002:** Characteristics of decoupling types between rural population and arable land.

Type	Number of Areas	Arable Land (%)	Settlements Area (%)	Population (%)
T3	14	64.01	58.69	59.79
T4	8	27.89	32.35	30.3
T5	3	8.1	8.96	9.91

**Table 3 ijerph-20-03833-t003:** Characteristics of decoupling types between rural population and rural settlements.

Type	Number of Areas	Arable Land (%)	Settlements Area (%)	Population (%)
T3	21	95.53	93.66	93.57
T4	3	4.22	5.47	4.33
T8	1	0.25	0.87	2.1

## Data Availability

The data presented in this study are available on request from the corresponding author.

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
