# Peer review of "Research on Rural Population/Arable Land/Rural Settlements Association Model and Coordinated Development Path: A Case Analysis of the Yellow River Basin (Henan Section)"

_ijerph, 2023, doi:10.3390/ijerph20053833_

Round 1
Reviewer 1 Report
The manuscript aimed to research the rural population-arable land-settlement association of the Yellow River Basin (Henan Section). However, it is needed to be major revised to be published. The comments are as follows.
1. The innovation of the manuscript mentioned was the correlation of people-farming-settlement. But there were no new findings with the analysis of the correlation. Please clarify the new findings.
2. The Discussion section should compare with other studies in 4.1 and 4.2, and describe the new findings in 4.3.
3. The minor revision. (1) Line 24, T3 in the abstract is confusing. (2) Line 96, the correlation between people-farming-settlement was not native. Line 231, 239, etc. (3) Line 177, the title of "3.1 Temporal and spatial changes of rural population-arable land-settlement" should be changed to "...changes of the rural population, arable land, and settlement". (4) line 188, the comma.
Author Response
General Comments: The manuscript aimed to research the rural population-arable land-settlement association of the Yellow River Basin (Henan Section). However, it is needed to be major revised to be published. The comments are as follows.
Response: Thank you for your timely and positive comments, our detailed point to point revision is as follows.
- The innovation of the manuscript mentioned was the correlation of people-farming-settlement. But there were no new findings with the analysis of the correlation. Please clarify the new findings.
Response #1:The temporal and spatial changes of rural population-cultivated land-rural residence show that in most areas, the permanent rural population decreases,cultivated land decreases, and the rural settlements increase. It is worth noting that the area of rural settlements has not decreased with the decrease of the number of permanent rural residents, on the contrary, the area of rural settlements has increased, which is a very worthy point in my research results, and it is also an important research focus to explore whether the changes in the time and space of rural population-arable land-rural living are coordinated or uncoordinated.
- The Discussion section should compare with other studies in 4.1 and 4.2, and describe the new findings in 4.3.
Response #2:Corrected.Please refer to 4.1,4.2 and 4.3
- The minor revision. (1) Line 24, T3 in the abstract is confusing. (2) Line 96, the correlation between people-farming-settlement was not native. Line 231, 239, etc. (3) Line 177, the title of "3.1 Temporal and spatial changes of rural population-arable land-settlement" should be changed to "...changes of the rural population, arable land, and settlement". (4) line 188, the comma.
Response #3: Corrected.
Reviewer 2 Report
The paper is well organized, while, I would have to say, it is an interesting and good research topic in China. However, the author hasn’t provided a very rigorous research analysis framework and mechanism analysis. Overall, the paper has some insights for the land-population-rural settlements harmony development. based on these, I ask the authors to check and modify the paper very seriously. My specific comments are in the bellowing:
(1) The abstract hasn’t been well organized, In the title you use “rural population—— arable land ——settlement” , while in the abstract, you use “rural people——plowing—settlement”, I think they are different, you need to keep consistence. Some language problems, such as “land use control and control”. These need to pay attention.
(2) In the abstract, you should simply put forward you core research question. Your research data and methodology, and the results and the conclusions, in the currents stage, the abstract is too long, and the results are too complex, which need to simply, the conclusion without some specific policy recommendations, also have not presented clearly what research methodology you are used.
(3) The authors have tried to do a good literature review and summarization which can helps to find a research gap of this paper, While, I am suspecting, as for each point you listed, it will not only one reference regarding on it, it should have one types of literature within a lot of scholars, otherwise, your literature reading is not enough.
(4) Normally, at the last of the introduction, you need to introduce the structure of the context, and introduce the arrangement of each section in the following paper.
(5) The major problems lie in that the author cannot well construct the relationship between the rural population-arable land-rural settlements. That is to say you haven’t constructed a good theoretical framework, such as you can use the human-land relationship theory in the geography to explain the changes in rural population may have influence on the land use and land structures, and also the human behaviors, then the rural settlements in the villages and rural areas, the authors haven’t constructed such a framework and then enter into the stage of empiric analysis and methodology.
(6) The formats problems in line 187-188 should be corrected. In line 216, people reducing land at the same time, this expression is horrible, need to be corrected.
(7) The Line spacing in section 3.2 and section 3.3 is different, these Academic normative issues should be pay much attention.
(8) I think the authors has lacked of a very important problem analysis, which is the influence mechanism analysis, as you have found the population change, the arable land changed and the rural settlements changes, also the coordination relationships between them, you should do a mechanism analysis, which factors has done great impacts on the changes of rural settlements, the land change, the rural population, the urbanization, the construction in the suburban areas……, these influence mechanism analysis will help you Deeping the analysis of the paper.
(9) The English expression is so poor, such as “the relationship between people and land has stages and complexity” The optimization path for coordinated development of rural population-arable land-settlement should put in the section 5, conclusion and policy recommendations, you should talk about them detailed in the policy recommendation, so you need to adjust the structure of the context.
(10) The authors always change the concepts/term “rural population——arable land -rural settlements” frequently, it is very bad, you should use consistent terms in the aper while only some little changes in the specific context, The authors haven’t put forward some valid policy recommendations in the conclusion section, and just present the conclusion, we need to see very specific and the corresponding policy tools that can promoted the rural harmony development from your research results, and the policy recommendation should also be actionable.

Author Response
The paper is well organized, while, I would have to say, it is an interesting and good research topic in China. However, the author hasn’t provided a very rigorous research analysis framework and mechanism analysis. Overall, the paper has some insights for the land-population-rural settlements harmony development. based on these, I ask the authors to check and modify the paper very seriously.
Response: Thank you for your timely and positive comments, our detailed point to point revision is as follows.
(1) The abstract hasn’t been well organized, In the title you use “rural population—— arable land ——settlement” , while in the abstract, you use “rural people——plowing—settlement”, I think they are different, you need to keep consistence. Some language problems, such as “land use control and control”. These need to pay attention.
Response #(1):Thank you for the suggestion, I have revised
(2) In the abstract, you should simply put forward you core research question. Your research data and methodology, and the results and the conclusions, in the currents stage, the abstract is too long, and the results are too complex, which need to simply, the conclusion without some specific policy recommendations, also have not presented clearly what research methodology you are used.
Response #(2):Thank you very much for your suggestion, I have revised it as requested, Please refer to abstract.
(3) The authors have tried to do a good literature review and summarization which can helps to find a research gap of this paper, While, I am suspecting, as for each point you listed, it will not only one reference regarding on it, it should have one types of literature within a lot of scholars, otherwise, your literature reading is not enough.
Response #(3):The references section has been adjusted and supplemented, and I must pay more attention to this in future writing.
(4) Normally, at the last of the introduction, you need to introduce the structure of the context, and introduce the arrangement of each section in the following paper.
Response #(4): There is a corresponding expression at the end of the introduction.
(5) The major problems lie in that the author cannot well construct the relationship between the rural population-arable land-rural settlements. That is to say you haven’t constructed a good theoretical framework, such as you can use the human-land relationship theory in the geography to explain the changes in rural population may have influence on the land use and land structures, and also the human behaviors, then the rural settlements in the villages and rural areas, the authors haven’t constructed such a framework and then enter into the stage of empiric analysis and methodology.
Response #(5): Thank you very much for putting forward such a good suggestion, establishing the corresponding framework, and then entering the stage of empirical analysis and methodology, which is a very scientific research idea, while the research in this paper mainly focuses on exploring the coordinated resettlement and optimization path of rural population-cultivated land and residential land. I will focus on framework construction and method research in future research. Thank you again for providing me with such a good research idea.
(6) The formats problems in line 187-188 should be corrected. In line 216, people reducing land at the same time, this expression is horrible, need to be corrected.
Response #(6):Thanks for your comment, corrected.
(7) The Line spacing in section 3.2 and section 3.3 is different, these Academic normative issues should be pay much attention.
Response #(7): It has been modified as you suggested. Please refer to section 3.2 and section 3.3
(8) I think the authors has lacked of a very important problem analysis, which is the influence mechanism analysis, as you have found the population change, the arable land changed and the rural settlements changes, also the coordination relationships between them, you should do a mechanism analysis, which factors has done great impacts on the changes of rural settlements, the land change, the rural population, the urbanization, the construction in the suburban areas……, these influence mechanism analysis will help you Deeping the analysis of the paper.
Response #(8):Thank you very much for your suggestion, which gave me a good idea for the study of human-land relations, and I used relevant theories in the discussion section to explain the reasons for the research results, thank you again.
(9) The English expression is so poor, such as “the relationship between people and land has stagesand complexity” The optimization path for coordinated development of rural population-arableland-settlement should put in the section5, conclusion and policy recommendations, you should talk about them detailed in the policy recommendation, so you need to adjust the structure of the context.
Response #(9):Thank you very much for your suggestions. As you suggested, sentences with grammatical problems have been revised, the optimization path is placed in the section5.The paper has been carefully revised by a professional language editing service to improve the grammar and readability.
(10) The authors always change the concepts/term “rural population——arable land -rural settlements” frequently, it is very bad, you should use consistent terms in the aper while only some little changes in the specific context, The authors haven’t put forward some valid policy recommendations in the conclusion section, and just present the conclusion, we need to see very specific and the corresponding policy tools that can promoted the rural harmony development from your research results, and the policy recommendation should also be actionable.
Response#(10): Thank you very much for your suggestions. Related terms have been harmonized. Correspondingpolicy recommendations have been added to the concluding section.
Round 2
Reviewer 1 Report
I don't think the authors took my comments seriously. Please revise the manuscript according to the comments, or make reasonable explanations for not revising with some comments.
Author Response
We are very sorry that the last revision did not meet your satisfaction. We have carefully revised the suggestions you put forward. Thank you for your timely and positive comments, our detailed point to point revision is as follows.
1.The innovation of the manuscript mentioned was the correlation of people-farming-settlement. But there were no new findings with the analysis of the correlation. Please clarify the new findings.
Response #1: Thank you for your advice. The new findings of this paper is mainly reflected in two aspects, and further elaborated in the paper. Please refer to 4.2 and 4.4.
(1) Previous studies mainly focused on the relationship between population and rural settlements. This paper carries out research from three aspects: population, arable land and rural settlements.
(2) According to the different decoupling states between rural population, arable land and rural settlements, differentiated optimization path are proposed, which is another innovation point of this paper.
2.The Discussion section should compare with other studies in 4.1 and 4.2, and describe the new findings in 4.3.
Response #2: Thank you very much for the revision suggestions. According to your suggestion, the discussion part was reorganized. The new findings are reflected in 4.2 and 4.4. In addition, the study of driving mechanism is added in the discussion part(This is also the suggestion of another reviewer.).
4.1 This part compares the Yellow River Basin (Henan section) with China's rural studies and finds that the research results tend to be consistent.
4.2 This part puts forward the innovation point of the paper. This paper innovates the research from the coordination relationship between rural population, arable land and rural settlements.
4.3 According to the uncoordinated development status of rural population-arable land-rural settlements in the Yellow River Basin (Henan section), targeted driving mechanism are put forward.
4.4 The coordinated development path of rural human-land relationship is put forward from two aspects.â‘ Optimization path of arable land reduction decoupling combination area.â‘¡Optimization path for development of rural population reduction and combined decoupling combined regional development.
3.The minor revision. (1) Line 24, T3 in the abstract is confusing. (2) Line 96, the correlation between people-farming-settlement was not native. Line 231, 239, etc. (3) Line 177, the title of "3.1 Temporal and spatial changes of rural population-arable land-settlement" should be changed to "...changes of the rural population, arable land, and settlement". (4) line 188, the comma.
Response #3: Thank you for your advice. I am sorry for this. It has been modified according to your suggestion.
In addition, we found colleagues whose mother tongue is English to further improve the language.
Thank you again for your valuable suggestions. In the future research, we will continue to deepen our research according to your suggestions.
Reviewer 2 Report
The paper is well organized, while, I can find that the authors have devoted great efforts to modify and revise the paper, and I also find that the paper has been greatly improved. My remaining comments that will help the authors to improve the quality are in the bellowing:
(1) The abstract has been well organized and improved, while, you should introduce the great topic first, such as “high-quality development” and “rural vitalization”, then the specific area “Henan”. Also, the policy suggestions, you should put forward in the abstract.
(2) I am quite admired that the authors have made great and improved literature review, it is a great job after the first-round comments, While I still think the authors lacks the theoretical framework, that means you need a theoretical framework to construct and link the whole paper, your empirical analysis result will follow the framework, it should be a framework diagram, which makes the readers very clear.
(3) The sentence in line 262 and 264 has some grammar problem, these problems cannot be forgiven in the scientific article.
(4) The sentence in line 306, 312, and 313, please carefully check the problems, I cannot accept these simple questions in the paper, I still think the paper lacking an influence mechanism analysis, if the author can reveal and illustrate the influence factors that have resulted in the different characteristics and decoupling states, it will be quite well in the scientific paper, and the research will be deepen.
(5) The policy implication in the end of the paper should be more specific and it should be simplify, focus and just talk about the specific measure. In addition, the limitation of the research should be put after the conclusion, which you can also write some research prospects.
Author Response
The paper is well organized, while, I can find that the authors have devoted great efforts to modify and revise the paper, and I also find that the paper has been greatly improved. My remaining comments that will help the authors to improve the quality are in the bellowing:
Response: Thank you for your timely and positive affirmation of my paper modification, our detailed point to point revision is as follows.
(1) The abstract has been well organized and improved, while, you should introduce the great topic first, such as “high-quality development” and “rural vitalization”, then the specific area “Henan”. Also, the policy suggestions, you should put forward in the abstract.
Response#1: Thank you for your advice. The abstract has been modified according to your suggestion. Please refer to line11-15 and line32-35.
(2) I am quite admired that the authors have made great and improved literature review, it is a great job after the first-round comments, While I still think the authors lacks the theoretical framework, that means you need a theoretical framework to construct and link the whole paper, your empirical analysis result will follow the framework, it should be a framework diagram, which makes the readers very clear.
Response#2: Thank you for your affirmation. The suggestion has greatly improved my paper. A theoretical framework has been added to the paper, Please refer to Fig1.
Figure 1. Theoretical framework diagram
(3) The sentence in line 262 and 264 has some grammar problem, these problems cannot be forgiven in the scientific article.
Response#3: Thank you for your advice. I am sorry for this. It has been modified according to your suggestion. Please refer to line 267-270.
In addition, we found colleagues whose mother tongue is English to further improve the language.
(4) The sentence in line 306, 312, and 313, please carefully check the problems, I cannot accept these simple questions in the paper, I still think the paper lacking an influence mechanism analysis, if the author can reveal and illustrate the influence factors that have resulted in the different characteristics and decoupling states, it will be quite well in the scientific paper, and the research will be deepen.
Response#4: Thank you for your advice. The analysis of driving mechanism is a very important aspect of this study. In this paper, the driving mechanism of the common uncoordinated state in the Yellow River Basin (Henan section) is qualitatively analysed. Please refer to 4.3.
Your advice provides a good idea for our research. In future research, we will further quantitatively analyze the driving mechanism by constructing the index system of influencing factors. Thank you again for your valuable advice.
(5) The policy implication in the end of the paper should be more specific and it should be simplify, focus and just talk about the specific measure. In addition, the limitation of the research should be put after the conclusion, which you can also write some research prospects.
Response#5: The discussion part was reorganized, and the optimization path was put into the discussion part, the limitation part was put into the conclusion part. At the same time, the prospect is added to the conclusion. Please refer to 4.4 and 5.2.
Thank you again for your valuable suggestions. In the future research, we will continue to deepen our research according to your suggestions.

Round 3
Reviewer 1 Report
I think the revised manuscript suits the journal.